# *Ginkgo biloba* L. Responds to Red and Blue Light: Via Phenylpropanoid and Flavonoid Biosynthesis Pathway

**Lei Zhang** , **Gaiping Wang \***, **Guibin Wang and Fuliang Cao**

Co-Innovation Center for Sustainable Forestry in Southern China, College of Forestry,
Nanjing Forestry University, Nanjing 210037, China; zlei@njfu.edu.cn (L.Z.);
gbwang@njfu.edu.cn (G.W.); fuliangcaonjfu@163.com (F.C.)
\* Correspondence: wanggaiping@njfu.edu.cn

**Abstract:** Light quality is a key environmental factor affecting plant growth and development. In this study, RNA-seq technology was used to explore the molecular mechanisms of ginkgo metabolism under different monochromatic lights. Leaves were used for transcriptome sequencing analysis after being irradiated by red, blue, and white LED lights. After treatment, 2040 differentially expressed genes (DEGs) were identified. Gene Ontology (GO) analysis showed that the DEGs were annotated into 49 terms. Kyoto Encyclopedia of Genes and Genomes (KEGG) enrichment analysis showed that 736 DEGs were enriched in 100 metabolic pathways, and 13 metabolic pathways were significantly enriched, especially 'phenylpropanoid biosynthesis' and 'flavonoid biosynthesis'. Further analysis of DEGs expression in the two pathways showed that *Ginkgo biloba* adapts to blue light mainly by promoting the expression of GbFLS to synthesize quercetin, kaempferol, and myncetin, and adapts to red light by promoting the expression of GbDFR to synthesize leucocyanidin. Nine DEGs were randomly selected for qRT-PCR verification, and the gene expression results were consistent with that of transcriptome sequencing. In conclusion, this study is the first to explore the molecular mechanism of ginkgo in response to different monochromatic lights, and it will lay a foundation for the research and application of light quality in the cultivation of leaf-use *G. biloba*.

**Keywords:** *Ginkgo biloba*; red and blue light; transcriptome; functional annotation; phenylpropanoid; flavonoid

## 1. Introduction

Light is the only energy source for organic matter synthesis during plant photosynthesis. It not only affects the biosynthesis and metabolism of chemical substances in plants [1] but also plays an irreplaceable role in regulating plant growth and development and light morphogenesis [2]. Light conditions are divided into three aspects: light cycle, light intensity, and light quality. Compared with the former two, the effect of light quality on plant growth and development is more complex [1]. Generally, red and blue lights are the two main spectra that drive photosynthesis and light morphogenesis of plants and, therefore, have the greatest impact on plants [3,4]. Studies have shown that, under different monochromatic lights, plants can selectively activate different photoreceptors to affect plant growth and development [5,6]. Phytochromes are red/far-red light sensors and are particularly prominent for their control of cell respiration and for facilitating the accumulation of plant organic matter, circadian rhythm, seed germination, and root development [6–9], whereas blue light plays an important role in the development of chloroplasts and the synthesis and accumulation of secondary metabolites through cryptochrome [10–14].

As a new type of artificial light source, light-emitting diodes (LEDs) can accurately and flexibly control the spectral range [15], and they have many advantages including low light weight, low energy consumption, reduced heat dissipation, long service life, and environmental protection [16]. LEDs have been widely used in the research, production, and cultivation of economic crops such as wasabi (*Wasabia japonica* L.) [17], tomato

(*Solanum lycopersicum* L.) [18], lettuce (*Lactuca sativa* L.) [19], strawberry (*Fragaria* × *ananassa* Duch.) [20], cucumber (*Cucumis sativus* L.) [21], and grape (*Vitis vinifera* L.) [22]; however, research on their application in forestry is rarely reported.

*G. biloba*, a native relic tree species in China, is naturally distributed in more than 20 provinces and cities. *G. biloba* is a natural treasure with many functions including medicinal [23–26], edible [27,28], ornamental [29,30], and for timber [31]. It has great ecological, economic, and research value [32]. Ginkgo leaves are rich in a variety of active substances such as flavonoids [33], terpenoids [34], phenolic acids [35], and polysaccharides [36]. In 2002, *G. biloba* products were included as raw food materials of the same origin as medicine and food in China. Related research on *G. biloba* leaves has been carried out for a long time, and research on its morphology and anatomy [37,38], growth physiology [39], medicinal value [40,41], content determination of active ingredients [42], and extraction technology [43,44] have become more in-depth and well-developed. Moreover, with the development of high-throughput sequencing, RNA-seq has been widely used in the field of bioscience. On the basis of previous growth physiology research, combined with RNA-seq and other technologies, this has become a hot topic in *G. biloba* research [45–47].

There are few studies on the effect of light quality on *G. biloba*. Only a few researchers have explored the changes in secondary metabolites of *G. biloba* under different lights by covering them with colored films [48,49]. In previous studies, the treatment methods were relatively rough, the control of light intensity, light cycle, and light quality were not accurate, and the research depth was shallow [48,49]. To solve these problems, LED technology was used to strictly control the lighting conditions, and *G. biloba* leaves were collected after culture for transcriptome sequencing. Through the methods of differential gene expression analysis and database function annotation, the metabolic forms of *G. biloba* responding to red and blue monochromatic lights were analyzed, and the functional genes responding to the different monochromatic lights were collated to provide a theoretical basis for studying the molecular mechanism of *G. biloba* growth under different monochromatic lights and lay to the foundation for the subsequent screening and verification of related functional genes. At the same time, this study provides theoretical support for the cultivation technology of leaf-use *G. biloba*.

## 2. Materials and Methods

### 2.1. Plant Materials and Treatments

On 21 October 2019, seeds were collected from ginkgo mother tree No. 16 in Xiashu Forest Farm, Nanjing Forestry University (118°58′–119°58′ E, 31°37′–32°19′ N) and stored in low-temperature sand. On May 12 in the following year, the seeds were soaked in clean water for 48 h, sown in a hole tray filled with clean wet sand, and placed in a light incubator for germination (16 h of light and 8 h of darkness, 25 °C), after which they were planted in non-woven containers and transferred to the artificial climate room of the biotechnology building of Nanjing Forestry University for cultivation. The substrate was peat–vermiculite–perlite (volume ratio 1:1:1). After three weeks of adaptation, seedlings with good growth and uniformity were selected for the light quality experiment.

Red (R) and blue (B) monochromatic lights were set, and white light (W) was used as a control (Figure 1). A completely randomized trial was conducted with 36 seedlings in each treatment group. The light source was an LED lamp (provided by Xiamen Sannonghui Photoelectric Technology Limited, Xiamen, China). The wavelength of blue light was 460–465 nm, the wavelength of red light was 660–665 nm, and the light intensity was 250 µmol m$^{-2}$ s$^{-1}$. The light cycle of the artificial climate chamber was 16 h of light and 8 h of darkness, and the culture temperature was 25 °C. After 2 months of different monochromatic light treatments, representative individual plants were randomly selected, and each treatment had three biological replicates. Young leaves were picked and placed into an RNase-free cryopreservation tube. After 30 min of freezing in liquid nitrogen, the leaves were stored at −80 °C prior to sample sequencing.

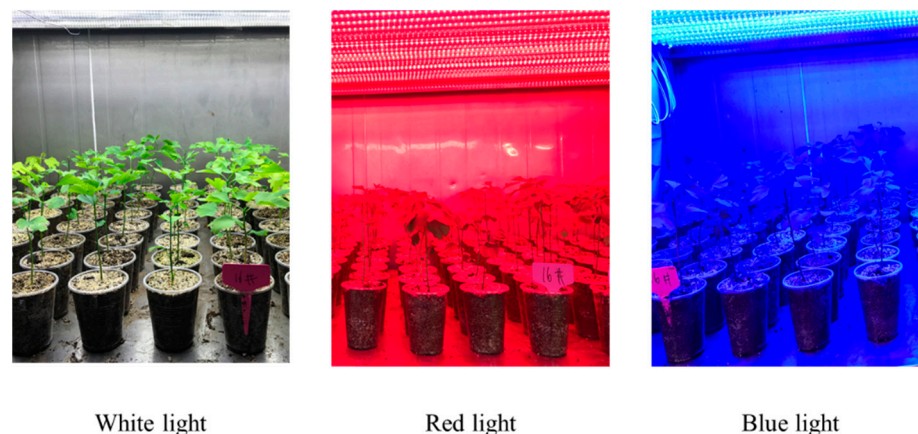

White light       Red light       Blue light

**Figure 1.** The growth of *G. biloba* seedlings under different monochromatic light.

### 2.2. Determination of Seeding Height, Diameter and Biomass

After different monochromatic light culture, 10 intact ginkgo seedlings were randomly selected for the determination of growth indicators. The height of seedlings was measured by tape measure [50,51], the ground diameter was measured by vernier caliper [50,51], and the biomass of roots, stems, and leaves was measured by electronic balance [52].

### 2.3. RNA Extraction, Library Construction, and Transcriptome Sequencing

RNA extraction, cDNA library construction, and sequencing were performed by Shanghai Meiji Biomedical Technology Co., Ltd. Total RNA was extracted using TRIzol (Invitrogen, Carlsbad, CA, USA). A Nanodrop 2000 (Thermo Fisher Scientific, Wilmington, DE, USA) was used to detect the mass and concentration of the extracted RNA. The integrity of the RNA was detected by agarose gel electrophoresis, and the quality of the RNA was checked using Agilent 2100 (value $\geq 7$). After passing the quality check, the RNA was sequenced using the Illumina Novaseq 6000 sequencing platform. Original Illumina sequencing data were deposited in the Sequence Read Archive (SRA) in National Center for Biotechnology Information (NCBI) with the BioProject ID: PRJNA723815.

### 2.4. Sequencing Quality Evaluation and Sequence Alignment

The clean reads obtained after quality control were compared with the reference genome of *G. biloba* (http://gigadb.org/dataset/100613, accessed on 4 June 2019) by HISAT2. The mapped data were used for subsequent transcript assembly, expression calculation, and alignment with six databases including NR (NCBI non-redundant protein sequences), Swiss-Prot (a manually annotated and reviewed protein sequence database), Pfam (Protein family), COG (Clusters of orthologous groups of proteins), GO (Gene Ontology), and KEGG (Kyoto Encyclopedia of Genes and Genomes). At the same time, the quality of the sequencing results was evaluated.

### 2.5. Screening, Functional Annotation, and Enrichment Analysis of Differentially Expressed Genes (DEGs)

The obtained clean reads were analyzed using the DESeq2 software. Based on the RPKM value, the screening conditions were $p$-adjusted $< 0.05$ and $|\log2FC| \geq 1$, and all DEGs were screened using pairwise comparison among the three groups, using COG and GO analyses, and KEGG enrichment analysis.

### 2.6. Quantitative Real-Time PCR (qRT-PCR) Validation of Differential Expression

In order to verify the results of transcriptome sequencing, according to KEGG enrichment results, 9 genes were randomly selected from phenylpropanoid biosynthesis and flavonoid biosynthesis pathways. Primer software (version 5.0) was used to design amplification primers. Glyceraldehyde-3-phosphate dehydrogenase (GAPDH) was used

as an internal reference gene for quantitative real-time fluorescence analysis. The specific primer sequences are listed in Table 1.

**Table 1.** The genes and primers used for qRT-PCR analysis.

| Gene ID | Forward Primer (5′–3′) | Reverse Primer (3′–5′) |
| --- | --- | --- |
| GAPDH | ATCCACGGGAGTATTCAC | CTCATTCACGCCAACAAC |
| Gb_23185 | CCTGGCCCATTCGCATT | AACCGAACTCCACACCTT |
| Gb_20355 | AAAAGTCCAAAGAAGCGGCAT | AACCAGGCCCAAACCCAA |
| Gb_07706 | AGCAAGGTAGCCATTCCA | AGCCATTGACAAATATGAGGA |
| Gb_29563 | CAGACCCCGGCACCATC | CTCCTTCTCCTCTATGTGCTT |
| Gb_10090 | GCTTCTCCCACAGGTCCG | CACAACCCAGAGATAGCCA |
| Gb_19792 | CCCTGCAAATCCATCCAC | ATTGCACAACATCAACGTC |
| Gb_14030 | CGCCCTCAAACCATACCTT | CTGCTTGCCAACCGTCT |
| Gb_10797 | TGCCCTCTCCCTCATCACC | CATAAACCCAAGATACCCGAT |
| Gb_10028 | GCTTATCCTACTGCCGGTCA | TTCGCACAACTCCGCTCT |

Total RNA was extracted using the same method as above, and then reverse transcribed into cDNA using the Aidlad TRUEscript RT kit, followed by qRT-PCR. The qRT-PCR reaction system (total 25 μL reaction) consisted of: 2 × SYBR Green Real-time PCR Master Mix (12.5 μL), positive and negative primers (0.5 μL each), cDNA template (1 μL), and water (10.5 μL). The reaction conditions were as follows: 95 °C for 15 s, 60 °C for 20 s, and 72 °C for 30 s, for a total of 40 cycles. The reaction was performed on an ABI 7500 Real-Time PCR system (Applied Biosystems, Foster City, CA, USA), with three biological replicates per sample. The relative expression of each gene was calculated by the $2^{-\Delta\Delta CT}$ method.

## 3. Results

### 3.1. Difference Analysis of Growth Indexes of G. biloba under Different Monochromatic Light

Under red and blue light, the growth of ginkgo seedlings changed significantly (Table S1). Compared with white light, the height and ground diameter of ginkgo seedlings decreased under red light. Under blue light, the ground diameter of *Ginkgo* decreased, but the seedling height increased significantly, reaching 20.42 cm, which was significantly higher than that under white and red light (Figure 2A). Compared with white light, the total biomass of *G. biloba* decreased under red and blue light. The root and stem biomass did not change significantly, but the leaf biomass decreased significantly (Figure 2B).

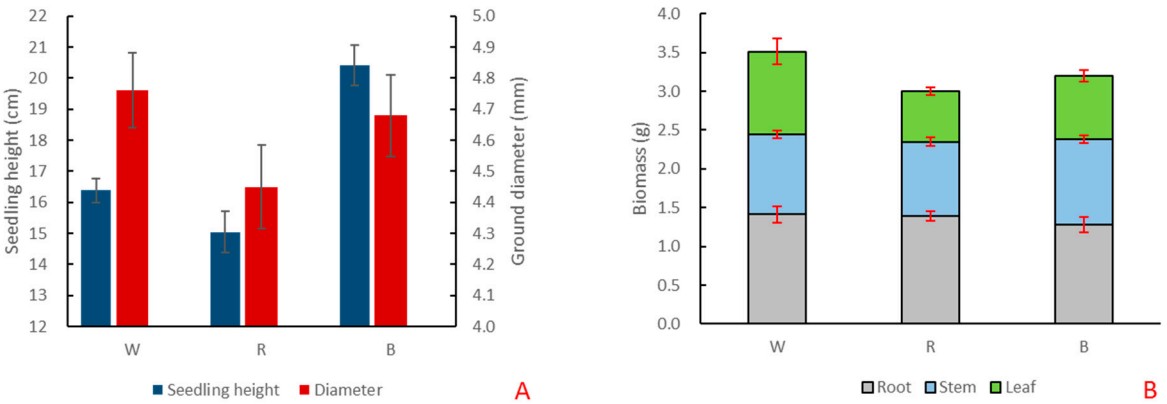

**Figure 2.** The growth changes of *G. biloba* under different monochromatic light. (**A**) Changes of seedling height and ground diameter of *G. biloba* seedlings under white (W), red (R), and blue (B) lights. (**B**) Biomass changes of roots, stems, and leaves of *G. biloba* seedlings.

### 3.2. Transcriptome Sequencing Statistics and Quality Assessment

Illumina sequencing technology was used to sequence nine cDNA libraries established from samples exposed to different monochromatic lights. After quality control, the average number of clean reads of *G. biloba* under white (W), red (R), and blue (B) lights was 50,880,797, 45,626,892, and 47,907,022, respectively. The clean reads of all samples were mapped to the reference genome of *G. biloba*, and the total mapping rate was above 93.42%. The unique mapping rate was between 86.50% and 89.22%, the quality score of the Q30 base was more than 93%, and the GC content was between 46.16% and 47.29%, indicating that the sequencing error rate was low, the overall quality was high, and that the reads could be used for subsequent data analysis. Detailed information on the sequencing is presented in Table 2.

**Table 2.** Statistics table of sequencing data.

| Sample | Raw Reads | Clean Reads | Total Mapped | Multiple Mapped | Uniquely Mapped | Q30 (%) | GC Content (%) |
|---|---|---|---|---|---|---|---|
| W1 | 50,260,172 | 49,857,286 | 47,146,280 (94.56%) | 4,020,380 (8.06%) | 43,125,900 (86.5%) | 95.48 | 47.29 |
| W2 | 49,152,842 | 48,745,716 | 45,997,218 (94.36%) | 3,420,423 (7.02%) | 42,576,795 (87.34%) | 95.16 | 46.65 |
| W3 | 54,486,246 | 54,039,390 | 51,176,322 (94.7%) | 4,387,848 (8.12%) | 46,788,474 (86.58%) | 95.09 | 47.2 |
| R1 | 51,370,906 | 50,981,158 | 48,424,745 (94.99%) | 3,615,814 (7.09%) | 44,808,931 (87.89%) | 95.29 | 46.76 |
| R2 | 44,458,876 | 43,839,486 | 40,955,359 (93.42%) | 2,795,455 (6.38%) | 38,159,904 (87.04%) | 93.19 | 46.02 |
| R3 | 50,901,456 | 50,503,512 | 47,500,571 (94.05%) | 3,419,144 (6.77%) | 44,081,427 (87.28%) | 94.99 | 46.43 |
| B1 | 50,958,110 | 50,583,964 | 48,020,218 (94.93%) | 2,888,507 (5.71%) | 45,131,711 (89.22%) | 95.28 | 46.55 |
| B2 | 44,497,382 | 44,157,342 | 41,615,152 (94.24%) | 2,775,224 (6.28%) | 38,839,928 (87.96%) | 95.19 | 46.16 |
| B3 | 49,391,368 | 48,979,760 | 46,078,100 (94.08%) | 3,361,108 (6.86%) | 42,716,992 (87.21%) | 95.02 | 46.82 |

Based on the expression quantity, principal component analysis was used for sample clustering analysis. The results showed that there was no outlier phenomenon in the sample, and the repeatability among the biological repeated samples was satisfactory (Figure 3).

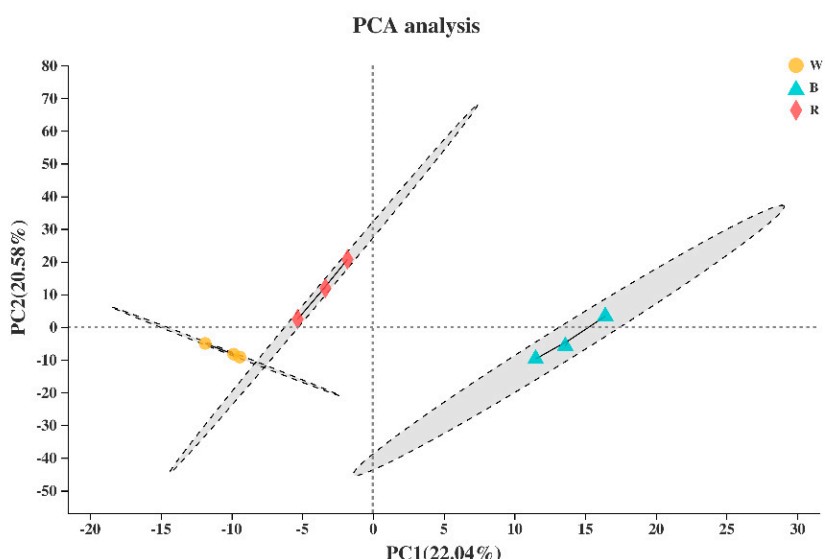

**Figure 3.** Principal component analysis (PCA) of the nine samples. W, B, and R represent white, blue, and red light, respectively.

### 3.3. Functional Annotation of Unigenes

All unigenes were compared with the GO, KEGG, COG, NR, Swiss-Prot, and Pfam databases (Figure 4). A total of 31,695 unigenes were annotated, and 23,678 (74.71%), 11,673 (36.83%), 25,155 (79.37%), 27,544 (86.9%), 21,961 (69.29%), and 18,677 (58.93%) unigenes were annotated in the above databases, respectively.

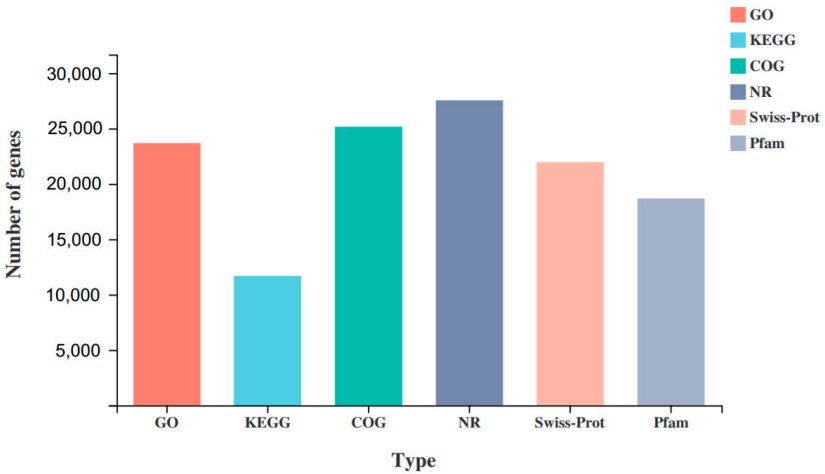

**Figure 4.** Functional annotation of unigenes.

## *3.4. Analysis of DEGs under Different Monochromatic Lights*

Based on the RPKM value, the differential expression of genes under different light treatments was analyzed. As shown in Figure 5A, 2040 DEGs identified from pairwise comparisons among the three groups of samples, and 21 genes were significantly different among the three groups. Among them, there were 655 DEGs between white light and red light, including 301 upregulated genes and 354 downregulated genes (Figure 5B), 1227 DEGs between white light and blue light, including 590 upregulated genes and 637 downregulated genes (Figure 5C), and 859 DEGs between red light and blue light, including 334 upregulated genes and 525 downregulated genes (Figure 5D).

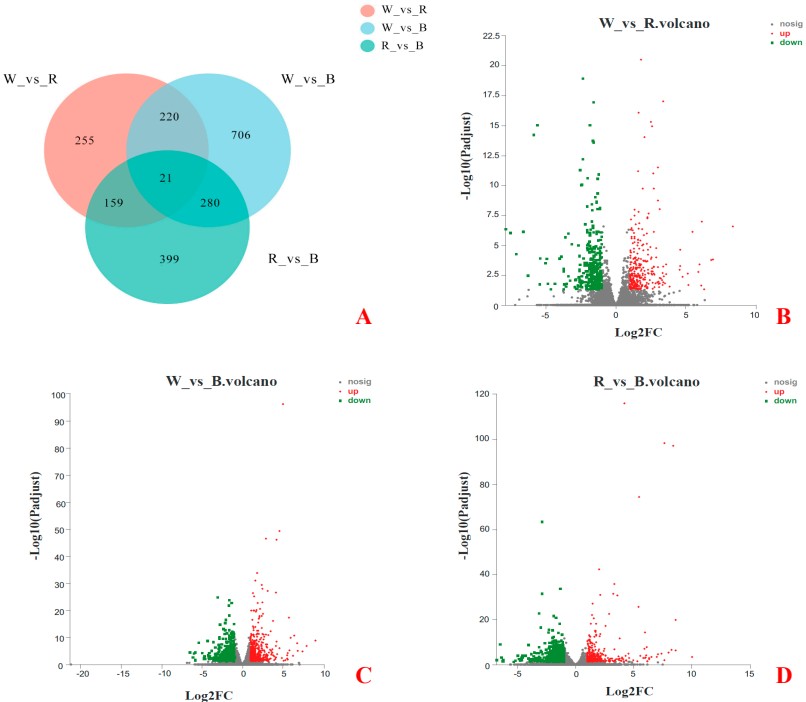

**Figure 5.** Differential gene expression among the three groups. (**A**) A Venn diagram describing overlaps among the DEGs in white (W), red (R), and blue (B) lights. (**B**) Volcano plot of DEGs between W and R lights. (**C**) Volcano plot of DEGs between W and B lights. (**D**) Volcano plot of DEGs between R and B lights.

### 3.5. Analysis of GO Annotations of DEGs

All DEGs in the three treatments were analyzed by GO analysis, and 2040 DEGs were annotated into 49 metabolic processes in the GO database. Among them, 22 DEGs were involved in 'biological process', 13 DEGs were involved in 'cellular component', and 14 DEGs were involved in 'molecular function'. The most annotated items were 'binding', 'catalytic activity', and 'membrane part', with 877, 869, and 706 annotated DEGs, respectively. The detailed functions of the 20 most abundant GO terms in the three categories are shown in Figure 6.

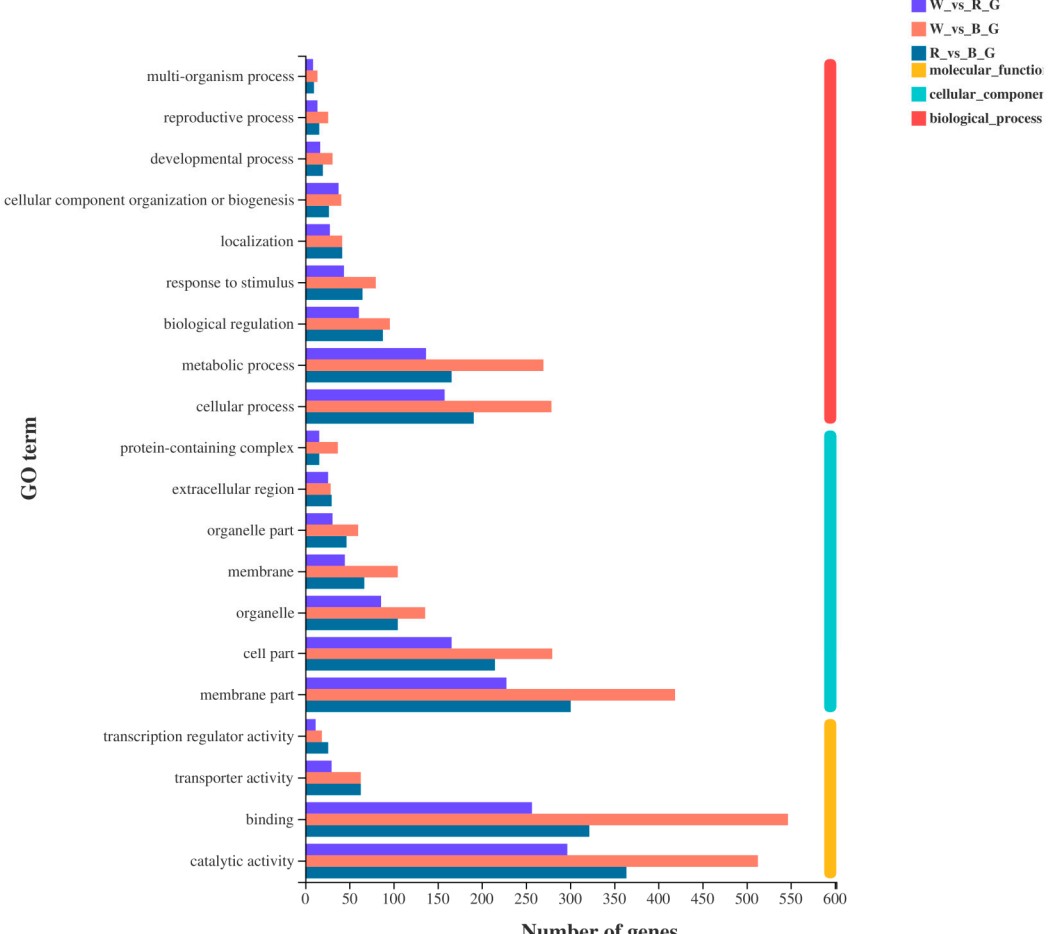

**Figure 6.** GO annotations analysis of DEGs.

Among them, the main biological process pathways were 'cellular process', 'metabolic process', 'biological regulation', 'response to stimulus', 'localization', 'cellular component organization or biogenesis', 'developmental process', 'reproductive process', and 'multi-organism process'. Cellular components included 'membrane part', 'cell part', 'organelle', 'membrane', 'organelle part', 'extracellular region', and 'protein-containing complex'. Molecular functions included 'catalytic activity', 'binding', 'transport activity', and 'transcription regulator activity'.

### 3.6. KEGG Enrichment Analysis of DEGs

To further explore the metabolic pathways and biological functions of ginkgo DEGs under different monochromatic lights, KEGG enrichment analysis was performed on all DEGs. The results showed that 736 DEGs were annotated in the KEGG database, with a total of 100 metabolic pathways and 13 significantly enriched pathways (Figure 7).

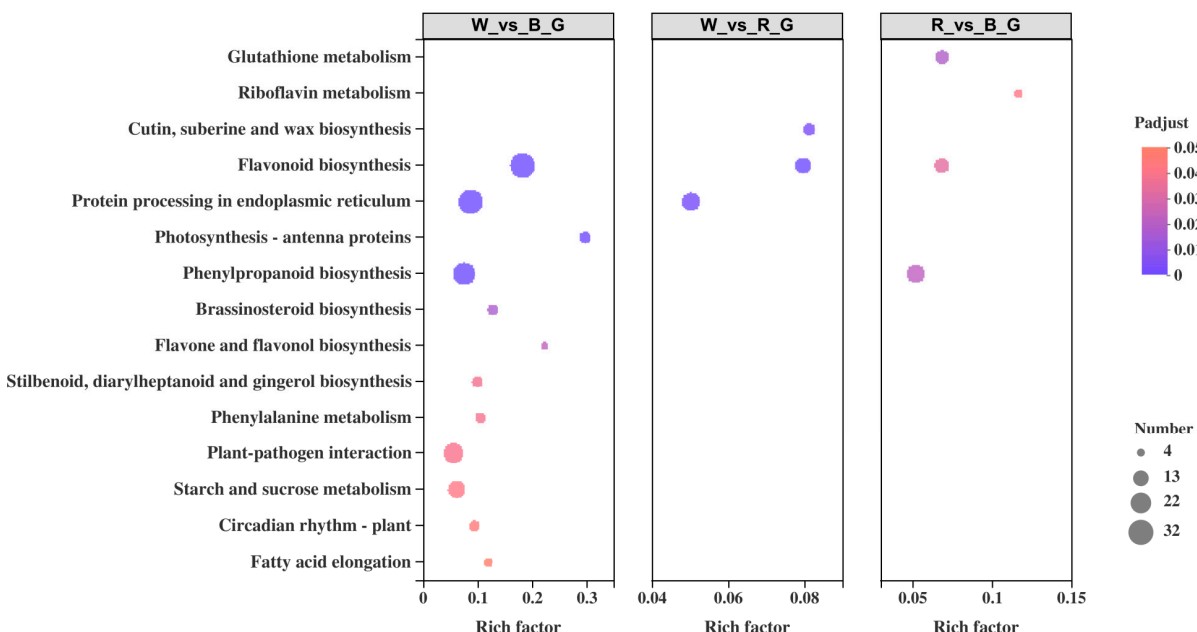

**Figure 7.** KEGG enrichment analysis of DEGs.

Among them, 12 metabolic pathways were significantly enriched between white light and blue light, including 'flavonoid biosynthesis', 'protein processing in endoplasmic reticulum', 'photosynthesis-antenna proteins', and 'phenylpropanoid biosynthesis'. Three metabolic pathways were significantly enriched between white light and red light: 'flavonoid biosynthesis', 'protein processing in endoplasmic reticulum', and 'cutin, suberine, and wax biosynthesis'. Four metabolic pathways were significantly enriched between red light and blue light: 'glutathione metabolism', 'phenylpropanoid biosynthesis', 'flavonoid biosynthesis', and 'riboflavin metabolism'. The most abundant DEGs were in 'flavonoid biosynthesis' and 'phenylpropanoid biosynthesis', with 41 DEGs in each pathway, followed by 'protein processing in endoplasmic reticulum' and 'plant-pathogen interaction', with 33 and 31 DEGs, respectively. The deepest enrichment was 'Photosynthesis-antenna proteins', followed by 'flavonoid biosynthesis'. It is worth noting that the pairwise comparison results all involved the process of 'flavonoid biosynthesis', in which 32 DEGs were enriched between white light and blue light, 14 DEGs were enriched between white light and red light, and 12 DEGs were enriched between red light and blue light.

### 3.7. Analysis of DEGs Related to the Phenylpropanoid Biosynthesis Pathway

Phenylpropanoid biosynthesis is an important pathway that links primary and secondary metabolism in plants. KEGG enrichment results showed that there were significant differences in the expression levels of 10 enzymes in the process of 'phenylpropanoid biosynthesis' under different lights, involving 41 related genes (Table S2). Among them, E1.11.1.7 had the largest number of DEGs, with a total of 14 DEGs, and the level of gene expression was high. Compared with white light, the gene expression of this enzyme was mostly upregulated under red light (except Gb_31015, Gb_20206, Gb_27962, Gb_14034), whereas the expression of the enzyme gene was mostly downregulated under blue light (except Gb_27962, Gb_41811, Gb_37299).

Three known key enzymes linked to secondary metabolism were selected to analyze the expression of related genes (Figure 8). Phenylalanine ammonia lyase (PAL), the first key enzyme and rate-limiting enzyme of phenylpropanoid biosynthesis, was upregulated under red and blue lights, and reached a significant level under blue light. In cinnamate-4-hydroxylase (C4H), the expression level of Gb_16449 increased under red light, which was significantly different from that under blue light. The expression of Gb_39990 in red light

was significantly lower than that in white light. Compared with white light, the expression of Gb_23185 increased under red and blue lights, and reached a significant level under blue light. In contrast to the former two enzymes, there was no significant difference in 4-coumaroyl-CoA ligase (4CL) gene expression.

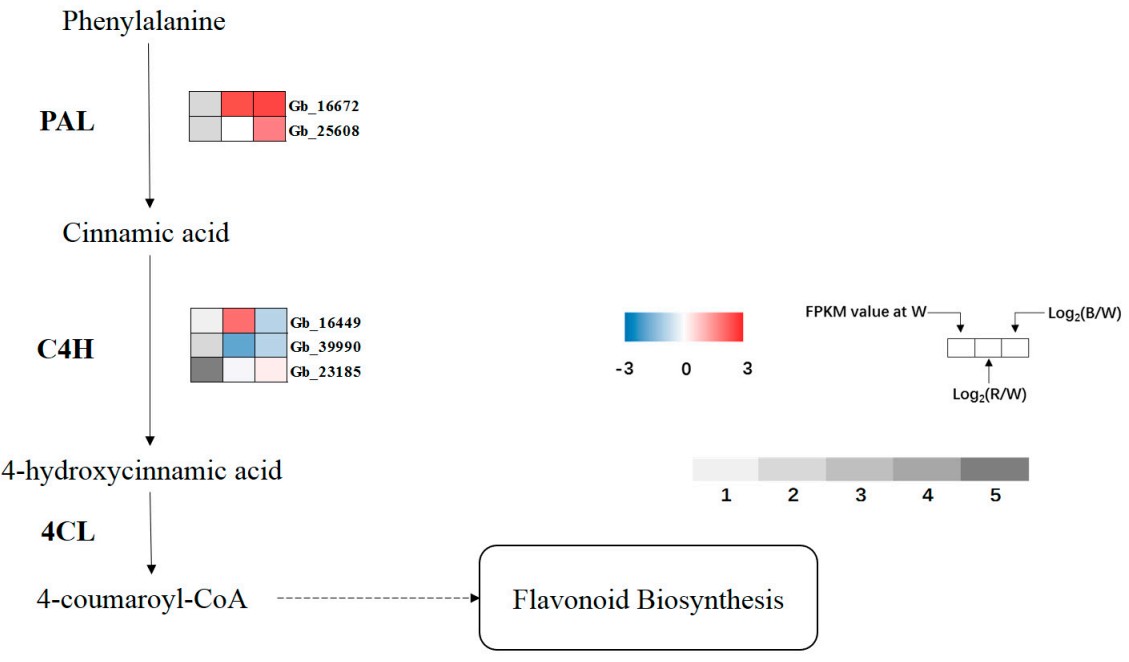

**Figure 8.** Expression profiles of DEGs involved in phenylpropanoid biosynthesis. Upregulated (red) and downregulated (blue) genes are shown. Grey bar represents FPKM value under white (W) light treatment, 1 represents FPKM values from 0 to 1, 2 represents FPKM values from 1 to 5, 3 represents FPKM values from 5 to 10, 4 represents values from 10 to 20, 5 represents values from 20 to 40.

### 3.8. Analysis of DEGs Related to the Flavonoid Biosynthesis Pathway

KEGG enrichment results showed that most DEGs were enriched in the flavonoid biosynthesis pathway after red and blue light treatment, and the degree of enrichment was very high. Further analysis of the specific pathway revealed a total of 13 flavonoid synthase enzymes (Table S3). Along with the synthesis process of ginkgo flavonol, a total of 10 key enzymes were selected to analyze the expression of related genes (Figure 9). In the whole metabolic pathway of ginkgo flavonol synthesis, the expression of enzyme genes increased significantly under blue light, whereas under red light, the synthesis of flavonol was regulated by the upregulation and downregulation of some genes.

Chalcone synthase (CHS) is a key enzyme that links phenylpropanoid metabolism and flavonoid biosynthesis. The expression of related regulatory genes increased significantly under blue light exposure. In addition to Gb_20355, expression of other regulatory genes increased significantly under red light. Flavonol synthase (FLS), an upstream enzyme that directly regulates the synthesis of kaempferol and quercetin, was significantly upregulated under blue light but only one gene (Gb_11130) was upregulated under red light, while other genes were significantly downregulated, and the Gb_14031 gene was not expressed. Since the upstream enzymes flavonoid 3′-hydroxylase (F3′H) and flavonoid 3′,5′-hydroxylase (F3′5′H) directly regulate quercetin synthesis (and regulate quercetin to myricetin transformation), the four genes were significantly upregulated under blue light, and their expression levels were also upregulated under red light but did not reach a significant level. The expression of dihydroflavonol reductase (DFR), anthocyanidin synthase (ANS), anthocyanidin reductase (ANR), and leucoanthocyanidin reductase (LAR) genes, which regulate the synthesis of anthocyanins, catechins, and other flavonoids, was

significantly altered. LAR was significantly downregulated under red and blue lights, while the expression of the other three enzymes (except Gb_26256 under blue light) was upregulated, and the expression of the ANS gene Gb_21868 was upregulated to the highest level under blue light.

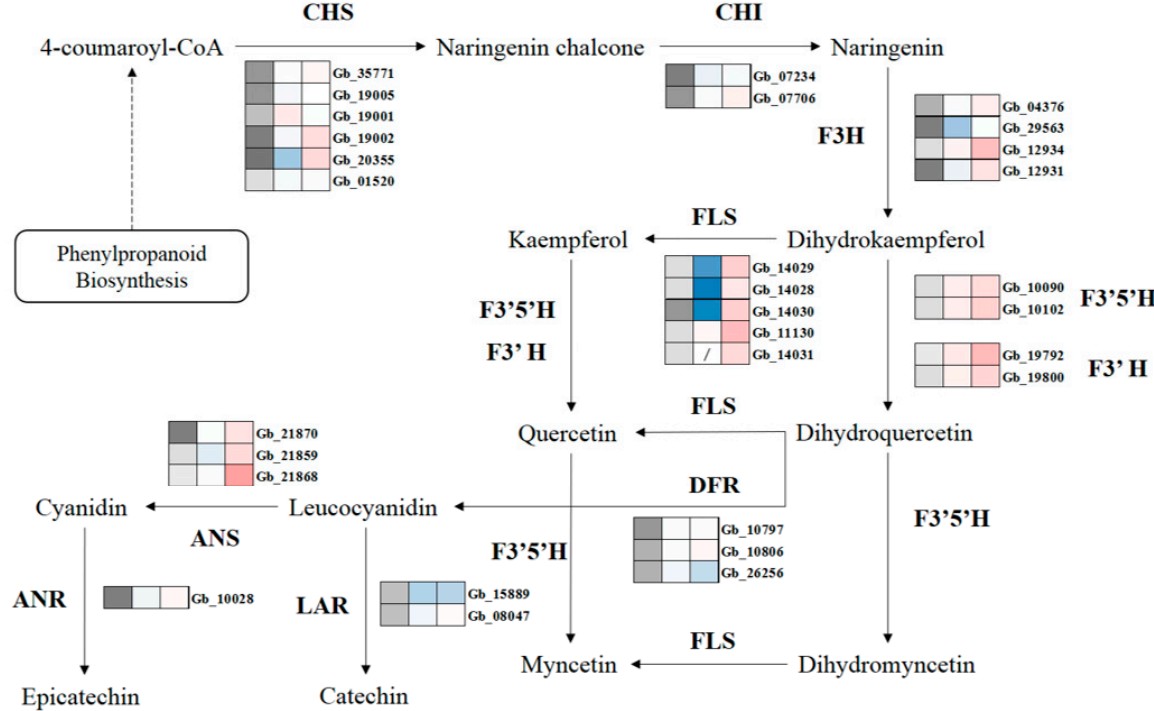

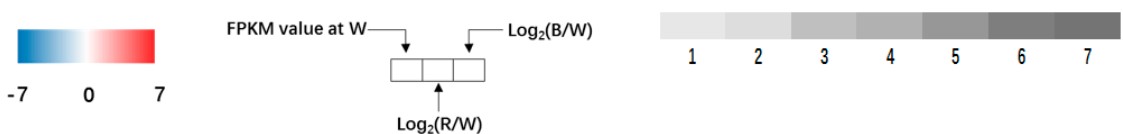

**Figure 9.** Expression profiles of DEGs involved in flavonoid biosynthesis. Upregulated (red) and downregulated (blue) genes are shown. Grey bar represents FPKM value under white (W) light treatment, 1 represents FPKM values from 0 to 1, 2 represents FPKM values from 1 to 5, 3 represents FPKM values from 5 to 10, 4 represents values from 10 to 20, 5 represents values from 20 to 40, 6 represents values from 40 to 80, and 7 represents values >80.

### 3.9. Validation of RNA-Seq Based DEGs Results by qRT-PCR

To verify the accuracy of RNA-Seq data, we randomly selected nine DEGs in the pathways of phenylpropanoid and flavonoid biosynthesis, using qRT-PCR. The qRT-PCR results showed that the expression trends of the nine enzyme genes under different monochromatic light were completely consistent with the trends of RNA-seq results (Figure 10).

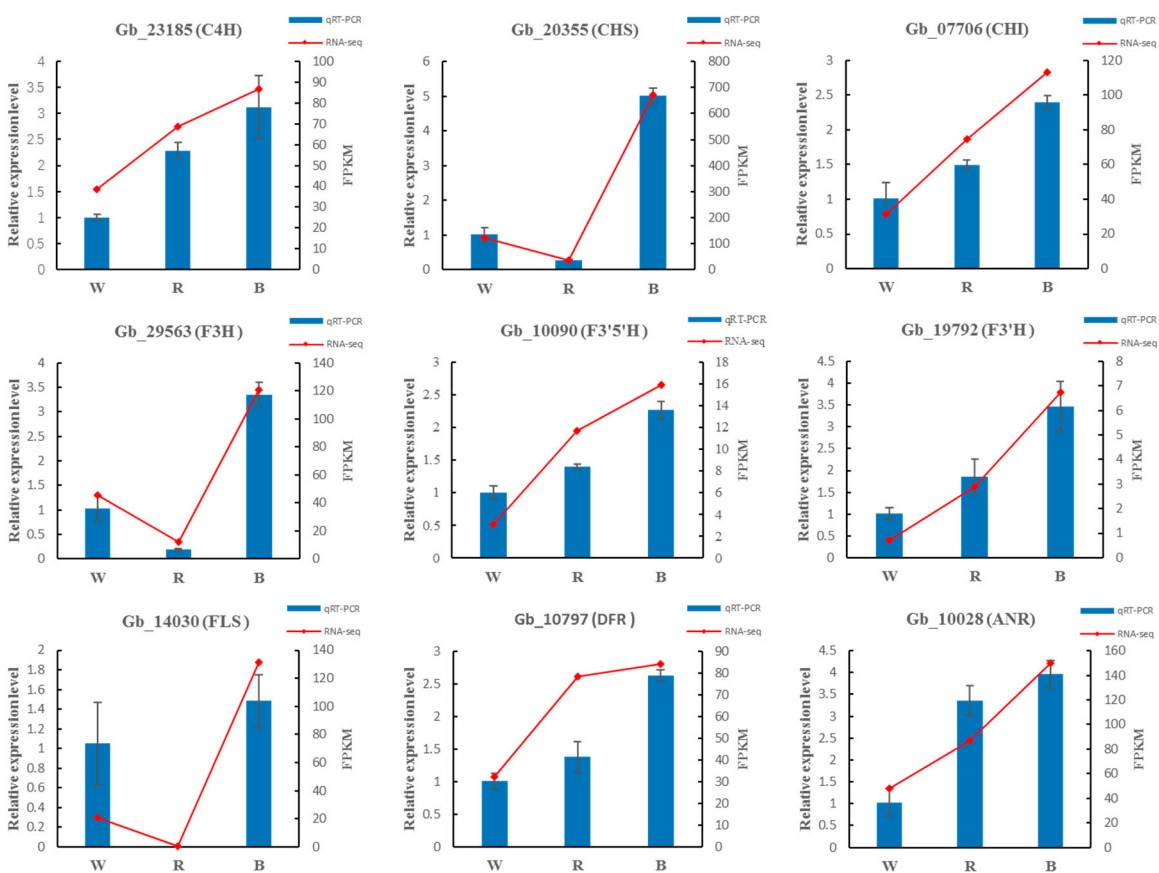

**Figure 10.** Validation of the RNA-Seq results by qRT-PCR.

## 4. Discussion

The response of *G. biloba* to different monochromatic lights is complex. This study revealed its molecular mechanism from the perspective of bioinformatics using high-throughput sequencing technology. The results showed that the stems of *G. biloba* seedlings were significantly elongated under blue light after 2 months, but neither blue or red light were conducive to the accumulation of leaf biomass. Leaves of *G. biloba* were selected for transcription sequencing. After quality control and comparison, 2040 differentially expressed genes were identified.

### 4.1. Functional Annotation Analysis of DEGs

The results of GO functional annotation analysis showed that the three most annotated subgroups of DEGs were related to 'binding', 'catalytic activity', and 'membrane part'. KEGG analysis showed that DEGs were mainly enriched in 'photosynthesis-antenna proteins', 'phenylpropanoid biosynthesis', and 'flavonoid biosynthesis'. The photosynthetic antenna protein is a key component of the light supplement complex for obtaining light energy [53]. After red and blue light irradiation, *G. biloba* leaves control the synthesis of their primary metabolites by changing the expression of genes related to photosynthesis-antenna proteins. Subsequently, the phenylpropanoid biosynthesis pathway and flavonoid biosynthesis pathway, which responded to red and blue light treatment, respectively, were used to realize the transition from primary to secondary metabolism.

### 4.2. Analysis of Phenylpropanoid Biosynthesis Pathway in G. biloba under Different Monochromatic Lights

Phenylpropanes are common substances in plants and play an important role in plant growth, development, and stress response [54,55]. The phenylpropanoid biosynthesis

pathway is one of the three major secondary metabolic pathways in plants, and regulates the synthesis of flavonoids, terpenoids, lignin, anthocyanins, and other secondary metabolites. It has been shown that the expression of genes related to the phenylpropanoid biosynthesis pathway can be activated by light [56]. The phenylpropanoid biosynthesis process is complex and involves a wide range of related enzymes. The three key upstream enzymes, PAL, C4H, and 4CL, play a major role in the whole process, catalyzing the formation of 4-Coumaroyl-CoA, a common precursor of the three secondary metabolic pathways.

PAL, as the first key enzyme in the phenylpropanoid biosynthesis pathway, connects primary metabolism with secondary metabolism and catalyzes the formation of cinnamic acid from L-phenylalanine. A previous study showed that PAL can regulate the normal growth and development of plants by participating in lignin synthesis [57]. In addition, PAL also regulates the synthesis of phytoalexin, flavonoids, phenols, and other substances, thus affecting the stress resistance of plants [58]. Therefore, PAL is used as a physiological marker of plant resistance [59]. Olsen et al. [60] reported that low temperatures can induce the overexpression of PAL enzyme genes, thus increasing the content of soluble phenylpropanoids, sinapic acid esters, and flavonoids for resistance against low temperatures. Heavy metals [61], hormones [62], and other stress treatments can also promote the expression of the PAL gene to achieve the effect of stress resistance. In the current study, compared with white light, two differentially expressed PAL genes (Gb_16672, Gb_25608) in *G. biloba* under monochromatic red and blue light were detected, which is similar to the past study. It may be that, under monochromatic red and blue light, *G. biloba* leaves can promote the synthesis and activity of the PAL enzyme by overexpression of the PAL enzyme gene, thus accelerating the secondary metabolism process of *G. biloba* and adapting to different monochromatic lights.

C4H and 4CL enzymes, the two enzymes downstream of the phenylpropanoid biosynthesis pathway, catalyze the formation of p-Coumaric acid and 4-Coumaroyl-CoA, respectively. The function of these enzymes is similar to that of PAL, through regulating the secondary metabolism process to allow plants to adapt to different stress environments [63]. The results showed that there were significant changes in C4H enzyme genes under the three lights, and there were significant differences in the expression levels of the three genes. Gb_39990 was downregulated under red and blue lights, and Gb_23185 was upregulated under red and blue lights, whereas Gb_16449 was upregulated under red light and downregulated under blue light, indicating that the gene may adapt to different monochromatic lights through different expression forms. At present, the 4CL gene has been identified in *Arabidopsis* [64] and rice [65]. Our sequencing results also showed that eight 4CL enzyme genes were annotated but the changes were not significant. In general, similar to other abiotic stresses, light quality can contribute to stress resistance by regulating some key enzyme genes in the phenylpropanoid biosynthesis process of *G. biloba*, and this has previously been verified in *Camptotheca acuminata* [56].

*4.3. Analysis of the Flavonoid Biosynthesis Pathway in G. biloba under Different Monochromatic Lights*

Flavonoids are important secondary metabolites in ginkgo leaves. They can deal with some abiotic stresses by activating antioxidant enzyme activity and scavenging free radicals [66]. The results of the present study show that flavonoid biosynthesis is the metabolic pathway with the highest concentration of DESs; 41 DEGs were enriched and 13 metabolic enzymes were involved.

Under blue light, except for Gb_26256 and Gb_15889, the expression levels of other genes related to flavonoid synthesis in *G. biloba* were upregulated. Under red light, in addition to FLS genes that directly regulate kaempferol synthesis, the expression levels of most enzyme genes also showed an upward trend. This indicates that *G. biloba* can improve the activity of flavonoid synthase by promoting the expression of genes related to flavonoid biosynthesis under red and blue lights, thereby increasing the flavonoid content in *G. biloba* leaves in response to monochromatic light stress. Liu et al. [12] also indicated that red and blue lights were conducive to the accumulation of flavonoids in pea sprouts,

which is consistent with the results of the present study. Xu et al. [67] found that salt stress in *Apocynum venetum L.* significantly promoted the gene expression of AvF3′H, AvF3H, and AVFLS, thereby increasing the contents of quercetin and kaempferol to resist salt stress. Zhao et al. [68] found that the expression levels of FLS (Gb_22751) and F3′H (Gb_19792, Gb_04545, Gb_11520) were significantly increased, the content of flavonoids in leaves was significantly increased, and the antioxidant capacity of flavonoids was significantly enhanced. In addition, previous studies on drought [69] and low temperature [70] showed similar findings, which further support the conclusion of the current study.

## 5. Conclusions

After two months of red and blue light treatment, the transcriptome of *G. biloba* leaves was sequenced to explore the related genes and metabolic pathways in response to different monochromatic lights. In general, there were obvious differences in gene expression under different monochromatic lights in *G. biloba*, and these differential genes were enriched in multiple metabolic pathways. Among them, phenylpropanoid biosynthesis and flavonoid biosynthesis were highly enriched, and the most abundant DEGs were identified. Through further analysis of these two pathways, combined with previous research results, it can be inferred that *G. biloba,* under monochromatic red and blue light stress and mainly through phenylpropanoid and flavonoid biosynthesis pathway-related enzyme gene expression, regulates the production of corresponding flavonoids to adapt to different monochromatic lights. These results laid a foundation for the molecular mechanism of *G. biloba* to monochromatic light response and provided theoretical support for the application of light quality in leaf-use *G. biloba* cultivation.

**Supplementary Materials:** The following are available online at https://www.mdpi.com/article/10.3390/f12081079/s1, Table S1: Analysis on growth difference of *Ginkgo biloba* under different monochromatic light, Table S2: Gene expression of phenylpropanoid biosynthesis pathway-related enzymes, Table S3: Gene expression of flavonoid biosynthesis pathway-related enzymes.

**Author Contributions:** Data curation, L.Z.; writing—original draft preparation, L.Z.; writing—review and editing, G.W. (Gaiping Wang); funding acquisition, G.W. (Guibin Wang) and F.C. All authors have read and agreed to the published version of the manuscript.

**Funding:** This research was funded by the National Key Research and Development Program of China, grant number 2017YFD0600700.

**Acknowledgments:** We thank Shanghai Meiji Biomedical Technology Co., Ltd. for helping with RNA-sequencing.

**Conflicts of Interest:** We declare that there is no conflict of interest.

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
