# Peer review of "Ginkgo biloba L. Responds to Red and Blue Light: Via Phenylpropanoid and Flavonoid Biosynthesis Pathway"

_forests, doi:10.3390/f12081079_

Round 1

Reviewer 1 Report

In the study, RNA-seq technology was used to explore the molecular mechanisms of ginkgo metabolism under different monochromatic lights. The research presented in the manuscript is up-to-date and relevant. The latest methods were used. The manuscript is just perfectly prepared, I have no comments for it.  

Author Response

Dear Reviewer:

Thank you for your comments concerning our manuscript entitled Ginkgo biloba L. responds to red and blue light: via phenylpropanoid and flavonoid biosynthesis pathway. Your comments on the manuscript have given us great recognition, which also makes us more confident to continue the relevant research in the future.

With best wishes.

Reviewer 2 Report

Dear Authors

the manuscript is totally acceptable but in figure 2 you need to do some corrections (details are in the attached file) and also add references for clarification of used methods (e.g. electronic balance).

yours

Author Response

Response to Reviewer 2 Comments

Dear reviewer

Thank you for your serious and professional comments on our manuscript. These comments have a good guiding significance for the modification and improvement of our paper. According to your suggestion, we have carefully revised the article and hope to get your approval.

Point 1: you need to use column shaped charts. using line charts when you didn't used W,R,B lights for the same plant is not appropriate and is wrong.

Response 1: Thanks for your valuable suggestion. Following to your advice, we have revised it in the revised manuscript.

Manuscript:

revised manuscript:

Point 2: you need add references for clarification of used methods (e.g. electronic balance).

Response 2: Thanks for your valuable suggestion. Following to your advice, we have added some references in the revised manuscript. For example, the measurement of seedling height diameter refers to references 50 and 51, and the determination method of biomass refers to reference 52.

Thank you again for your professional and careful review.

With best wishes.

Reviewer 3 Report

The manuscript has interesting data about the role of LED light of different types on ginkgo growth and gene expression. However, it is well known for a long time that light affects secondary metabolism, in particular the phenylpropanoid one. Since no proteomic or physiological assays support the transcriptomic data the soundness of the paper is limited. Before ir can be accepted the authors must address the comments and corrections made in the edited manuscript I've uploaded. In particular, they must pay attention to the quality of some figures and the way they wrote the references.

Round 2

Reviewer 3 Report

The authors changed the original version according to the suggestions, so the ms can now be accepted for publication